# Detecting Diabetic Retinal Neuropathy Using Fundus Perimetry

**DOI:** 10.3390/ijms221910726

**Published:** 2021-10-03

**Authors:** Takayuki Baba

**Affiliations:** Department of Ophthalmology and Visual Science, Chiba University Graduate School of Medicine, Inohana 1-8-1, Chuo-ku, Chiba 260-8670, Japan; t.baba.oph@faculty.chiba-u.jp; Tel.: +81-43-226-2124; Fax: +81-43-224-4162

**Keywords:** diabetic maculopathy, fundus perimetry, mesopic, optical coherence tomography, retinal sensitivity, scotopic

## Abstract

Fundus perimetry is a new technique for evaluating the light sense in the retina in a point-to-point manner. Light sense is fundamentally different from visual acuity, which measures the threshold for discriminating and perceiving two points or lines, called the minimum cognoscible. The quality of measurement of retinal sensitivity has dramatically increased in the last decade, and the use of fundus perimetry is now gaining popularity. The latest model of fundus perimetry, MP-3, can be used for a wide range of measurements and has an advanced eye tracking system. High background illumination enables accurate measurement of mesopic retail sensitivity. Recent investigations have shown that neuronal damage precedes vascular abnormalities in diabetic retinopathy. The loss of retinal function has also been reported prior to morphological changes in the retina. In this review, the importance of measuring retinal sensitivity to evaluate visual function in the early stages of diabetic retinopathy was discussed. The usefulness of retinal sensitivity as an outcome measure in clinical trials for treatment modalities is also presented. The importance of fundus perimetry is promising and should be considered by both diabetes researchers and clinical ophthalmologists.

## 1. Rationale of Fundus Perimetry and Macular Disease

### 1.1. Visual Field and Retinal Sensitivity

The visual field is the area that can be seen without moving our eyes. The standard visual field is 50° upward, 60° nasally, 70° downward, and 90° temporally in humans [1]. Even though we have this range of vision, we can only see details at the center of our field of vision, with less clear vision in our periphery. Thus, the perception of vision differs depending on the visual field. Retinal sensitivity is the way we perceive light, and the visual field is an accumulation of regions with various retinal sensitivities which is measured by the difference between the background luminance and stimulus intensity. Typically, the center of the visual field is the most sensitive, and the sensitivity decreases as it moves to the periphery of the visual field. The “visual field island” is an easy-to-understand model of this sensitivity [1]. The land area of the island represents the sensitivity of the field of view, a higher island represents higher retinal sensitivity, and a lower land area at the edge of the island indicates lower retinal sensitivity. At a distance of approximately 15° temporal from the center, there is an area with zero retinal sensitivity called the blind spot that is approximately 5° in diameter [2]. This blind spot corresponds to the area where the optic nerve exits the eye, and since there are no light sensitive photoreceptor cells in this area (optic nerve papilla), it is an absolute dark spot.

### 1.2. What Is the Difference between Fundus Perimetry and Visual Acuity?

The first thing that comes to mind when evaluating visual function is visual acuity. In routine testing, the examinee tells the clinician which side the letter C (Landolt ring) appears open. This measures the threshold for discriminating and perceiving two points or two lines called the minimum cognoscible. In measuring visual acuity, the minimum value of this minimum cognoscible is used to determine the visual acuity of the patient. Visual acuity is highest at the center of the retina, where normal vision is 20/18 or better in Snellen acuity. In the human fovea, approximately 0.05° of visual space (three minutes of arc) is represented in 1 mm of striate of the cortex, while at 10° eccentricity, approximately 0.67° is represented in 1 mm. Human eyesight has a higher resolution in the center and a lower resolution in the periphery.

Retinal sensitivity, measured in visual field testing, is the ability to perceive the difference between the background luminance and stimulus intensity. Just as the letter “C” becomes smaller and smaller at the bottom of a visual acuity chart, the brightness of the stimulus presented during a visual field test to measure retinal sensitivity becomes darker and darker. In other words, if the examinee can perceive dark stimulus, the examinee has high retinal sensitivity and high visual function. In this way, visual acuity testing and visual field testing measure different visual functions, even though they are both included in visual testing.

### 1.3. Why Examine Retinal Sensitivity?

The width and depth of the visual field can be evaluated by measuring retinal sensitivity. There are two methods of measuring visual field testing. One of these is kinetic visual field testing, which measures the extent of the visual field by moving the stimulus. Goldmann perimetry is used as kinetic perimetry; the larger and brighter the stimulus, which can be seen even in the periphery of the visual field, the larger the visual field [2]. In contrast, the smaller the stimulus, the smaller the visual field. When contour lines connect this area (iso-sensitivity curve), a two-dimensional projection of the island, as mentioned above, of the visual field is created, and the range of these contour lines becomes narrower when there is damage to the retina or optic nerve. Other perimetry methods, such as static visual field testing using a Humphrey visual field analyzer or Octopus static perimeter, are also commonly used [3,4]. Automated static perimetry is commonly used to measure retinal sensitivity near the center of the visual field and is mainly used to evaluate glaucoma and optic nerve diseases. These visual field tests are subjective tests in which the examinee repeatedly presses a switch when they see a stimulus.

### 1.4. What Is the Difference between Fundus Perimetry and a Conventional Visual Field Test?

As mentioned above, the visual field consists of numerous measurements of individual retinal sensitivities, so each measurement point in the visual field corresponds to the retina in the fundus. Conventional visual field tests (kinetic and static visual field tests) can present the relative distribution of retinal sensitivity in the visual field at the center and periphery. However, they cannot show a one-to-one correspondence with the measurement position and retinal sensitivity. During a fundus perimetry test, a fundus monitor can show which retinal area is being stimulated by the stimulus in a timely manner [5,6]. The test result is a combination of positional information on the retina and retinal sensitivity. However, it does not have the same measurement range as a static visual field meter and cannot accurately match the retinal function for comparison [7,8,9].

### 1.5. Measurement Principle of Fundus Perimetry

While observing the fundus with a scanning laser ophthalmoscope (SLO) image, retinal function at a specific point on the retina is measured by tracking the movement of the eye. The localization of the stimulated retinal area may not be precise throughout the session because the fixation is not always stable. The mean eye tracking accuracy of 4.9 min of arc has been reported by MP-1 [10]. The tracking frequency is one of factors which varies from device to device. In addition, the patients with macular disease have poorer fixation stability with less accurate localization of stimulus. The head movements as well as the eye movement also affect the alignment during the test. By gradually changing the brightness of the stimulus from bright to dark, the darkest luminance (threshold) at which the stimulus can be differentiated from the background luminance is determined and is used as the retinal sensitivity [11]. This is a subjective test in which the examinee presses a switch when the stimulus is seen. The brightness of the stimulus is increased or decreased depending on the program. However, a method such as the 4-2-1 staircase is commonly used, in which the brightness is gradually reduced and narrowed until the threshold is determined. At the end of the examination, a fundus photograph is taken and superimposed on the SLO image, which is used during the measurement to display the retinal sensitivity on the actual fundus image (Figure 1).

### 1.6. Relationship between the Measurement of Retinal Sensitivity and SLO

During the measurement of retinal sensitivity with fundus perimetry, the fundus is observed using an SLO. If a normal fundus is examined using white light, scattered light irradiates the retinal surface in a planar manner, stimulating photoreceptor cells, making it difficult to measure retinal sensitivity accurately. SLO scans the fundus at high speed using a small, low-scattering light beam [12,13]. The laser wavelength is set in the infrared region. With an SLO, the amount of light required for fundus observation is only 1/100 to 1/1000 of that required for a normal fundus examination. If the laser wavelength is set to the infrared region and the stimulus is bright (current fundus perimetry uses LEDs), the photophobia of the examinee is eliminated, and the retinal sensitivity can be measured correctly. While referring to the fundus image obtained by SLO, fine movements of the eyeball can be detected. Moreover, by correcting the position of the stimulus by eye tracking, the retinal sensitivity can be accurately measured in line with the retinal area [14,15].

### 1.7. Types of Fundus Perimetry

A list of the fundus visual field meters currently in use is presented in Table 1. MP-1 (NIDEK, Aichi, Japan) fundus perimetry displays the stimulus on an LCD. The dynamic range of retinal sensitivity that could be measured is relatively narrow, around 0–20 dB. Later, Maia (MAcular Integrity Assessment, Centervue, Padova, Italy) was developed, which uses an SLO for fundus observation, and an optotype displayed on LED. The latest model of fundus perimetry, MP-3 (NIDEK, Aichi, Japan), has a measurement range of 0–34 dB. Another feature of the MP-3 is that the background luminance can be set as high as 31.4 asb, which enables the measurement of retinal sensitivity in photopic vision. This enables the measurement of retinal sensitivity to be conducted under similar conditions to everyday life [16,17].

### 1.8. Difference between Dark and Bright Fields

With MP-3, retinal sensitivity can be measured in photopic vision, making it possible to measure retinal sensitivity under similar conditions to everyday life. After 30 min of dark adaptation, the S-Maia (Centervue) displays the stimulus using two types of LEDs, cyan (505 nm) and red (627 nm), and measures the scotopic retinal sensitivity [18,19,20]. Cyan light can measure retinal sensitivity mainly derived from rods, while red light can measure retinal sensitivity mixed with rods and cones. Although slightly different from the retinal function used in everyday life, such dark vision retinal sensitivity, can detect retinal sensitivity loss more significantly than dusk vision retinal sensitivity in eyes with reticular pseudo-drusen, suggesting that it may help detect the early stage of pathologic lesions [21].

### 1.9. Retinal Sensitivity in Normal and Diseased Eyes

The retinal sensitivity of MP-3 fundus microperimetry in the normal eye is shown in Figure 1. The retinal sensitivity ranges from 22 to 36 dB in normal eyes and is displayed in green on a color map. The normal retinal sensitivity ranges from 24 to 36 dB in Maia, and from 12 to 20 dB in MP-1. If most of the area is displayed in green, it is considered normal. The light blue dot in the center indicates the fixation location, and if the fixation is unstable (i.e., the eye is moving during a test), the light blue dot will be scattered. When vision is good, and fixation is stable, the light blue dots concentrate on a single point. If the fixation is unstable, the measurement results are considered unreliable. There is also a function to detect whether the examinee is pressing the switch unintentionally using a false positive and false negative test. The red circle in the center is the fixation target displayed during the test, which can change its size and shape depending on the disease being tested [22]. Large, clear fixation targets are helpful in cases with poor central visual acuity. However, care must be taken because retinal sensitivity is significantly reduced when the stimulus overlaps with the fixation target.

## 2. Role of Fundus Perimetry to Detect Early Functional Loss in Diabetic Retinopathy

### 2.1. Diabetic Retinopathy

Diabetic retinopathy is a potentially blinding disease, but the symptoms are almost absent in the first ten years of diabetes mellitus [23]. In other words, diabetic retinopathy is silent in the early stage, and once the disease goes beyond the threshold, the patients first realize visual impairment. Regular follow-up of the fundus is strongly recommended to avoid occult progression of retinopathy. However, the number of patients who undergo regular check-ups is limited, and it is much worse in areas where accessibility to ophthalmology clinics is limited. To save the eyes from advancing to proliferative retinopathy, early detection of retinopathy is necessary [24]. Mass screening, such as residential health checks, is useful. However, fundus photography is not sensitive enough to detect early changes in diabetic retinopathy. Previous studies have revealed alterations in electroretinogram, contrast sensitivity, color vision, and retinal sensitivity results by fundus perimetry [25,26,27,28,29]. More recently, new modalities, including optical coherence tomography angiography, revealed early changes in the microvasculature of diabetic retinopathy when there was no apparent retinal change [30,31].

Nittala et al. reported reduced retinal sensitivity in the retina of diabetic patients without diabetic retinopathy [32]. The retinal thickness is also reduced in eyes with no apparent diabetic retinopathy [33,34,35,36].

In diabetic retinopathy, retinal edema due to leakage occurs because of damage to the retinal capillaries [37]. In particular, retinal edema often occurs in the macula and is an important cause of decline in visual function. If macular edema involves the fovea, central vision is significantly reduced, but central vision may be relatively preserved if it does not involve the fovea. Even in such cases, measurement of the retinal sensitivity with a fundus visual field meter shows a decrease in retinal sensitivity, consistent with macular edema (Figure 2). This decrease in retinal sensitivity may be recovered as retinal edema improves with treatment, such as intravitreal administration of anti-VEGF drugs and macular laser [38,39,40,41].

### 2.2. Functional Macular Changes before any Morphological Changes Occur in Patients with Diabetes

Recent studies have shown that changes in the retinal neurons occur prior to microvascular changes in diabetic retinopathy. The inner retina, including the ganglion cell complex (GCC) and inner plexiform layer, are already damaged in diabetic patients without diabetic retinopathy.

Verma et al. first compared 39 eyes of healthy patients with 39 eyes of diabetic patients without diabetic retinopathy using spectral-domain optical coherence tomography (SD-OCT) and fundus microperimetry [33]. The mean ages of each group were 50.9 years and 49.9 years in diabetic cases and healthy controls. The mean foveal thickness measured by SD-OCT was 168 and 178 micrometers in diabetic cases and controls, respectively. The mean thickness of the photoreceptor was also lower in patients with diabetes. Retinal sensitivity was measured by fundus perimetry (MP-1) in the central 20°. The test was performed at 33 points with a background illumination of 4 abs. The average retinal sensitivity was 15.7 dB (normal: 12–20 dB) and 17.7 dB in diabetic cases and controls, respectively. This age-matched case-control study suggested that early retinal thinning of the photoreceptor layer and reduced retinal sensitivity in the macular area occurred. Neurodegeneration precedes the development of diabetic retinopathy, and retinal function is reduced without a reduction in the central visual acuity. They also reported an association between a longer duration of diabetes and a thinner layer of photoreceptors. However, there was no association between the duration of diabetes, foveal thickness, and retinal sensitivity.

The same scholars reported a difference in the thickness of the nerve fiber layer and fundus perimetry in patients with and without diabetes. Seventy patients with diabetes without diabetic retinopathy were compared with 40 healthy controls [33]. MP-1 fundus perimetry measured retinal sensitivity in the central 20° with a background illumination of 4 abs. The retinal nerve fiber layer (RNFL) thickness was significantly lower in the eyes of patients with diabetes. The foveal sensitivity and retinal sensitivity of the central 20° were comparable. The duration and control of diabetes did not affect the retinal thickness, but the cases with glycosylated hemoglobin less than 7% had lower retinal sensitivity (14.1 vs. 15.4 dB (normal: 12–20 dB)). The authors did not mention the reason for this finding, and further studies are necessary.

Nittara et al. also reported reduced foveal sensitivity in diabetic patients without diabetic retinopathy [32]. They included 40 controls, 40 eyes with diabetes but no diabetic retinopathy, and 130 eyes with diabetic retinopathy. In eyes with diabetic retinopathy, 40 mild, 30 moderate, and 30 severe eyes with non-proliferative diabetic retinopathy and 30 with proliferative diabetic retinopathy were included. MP-1 measured the retinal sensitivity in the central 20° at 33 stimuli points. The mean foveal sensitivity in the central 2° was 16.7 dB (normal: 12–20 dB) in the control, 14.7 dB in cases with diabetes but no diabetic retinopathy, and 11.6 dB in participants with diabetic retinopathy. The mean retinal sensitivity (central 20° of the macula) was 17.6 dB in the control, 16.2 dB in participants with diabetes but no diabetic retinopathy, and 13.0 dB in participants with diabetic retinopathy. The eyes with severe diabetic retinopathy showed lower mean retinal sensitivity in the central 2 and 20°. This trend was confirmed in all four quadrants if retinal sensitivity was compared in each quadrant. This study suggests a subclinical reduction in retinal sensitivity in eyes with diabetes but no diabetic retinopathy.

Neriyanuri et al. investigated a cohort of patients with type 2 diabetes and diabetic neuropathy but not diabetic retinopathy [36]. They included 185 patients with diabetic neuropathy and 558 patients without neuropathy. The RNFL thickness measured by SD-OCT was significantly thinner in eyes with diabetic neuropathy, even though the patients did not have diabetic retinopathy. The retinal pigment epithelium thickness was also lower in the eyes with diabetic neuropathy. The mean retinal sensitivity measured by MP-1 fundus perimetry was 14.8 dB (normal: 12–20 dB) in eyes without neuropathy and 13.6 dB with neuropathy. There was a significant association between the mean retinal sensitivity and diabetic neuropathy. They suggested that early inner retinal damage in eyes with diabetic neuropathy resulted in impaired retinal sensitivity despite there being no diabetic retinopathy. Neuronal damage occurs prior to the development of microvascular damage.

A structure–functional analysis in early diabetic retinopathy was performed by Montesano et al. [35], who used SD-OCT to measure the retinal thickness and fundus perimetry to measure the retinal sensitivity. A total of 68 eyes from 35 healthy subjects and 48 eyes without diabetic retinopathy from 26 subjects with type 2 diabetes were included. They used Maia for the fundus perimetry, and the sensitivities were measured at 37 points in the central 10° with a background illumination of 4 abs. The pointwise relationship between the inner retinal thicknesses was significant at some points in the innermost ring (1°) closest to the central fovea. This relationship was not observed in healthy controls. The thicknesses of the inner retina were thinner in diabetic patients without diabetic retinopathy. Montesano et al. [35] suggested that this slight change indicated an early alteration of the neural retina in patients with diabetes. No differences in the central visual acuity and global analysis of the retinal sensitivity at the grid area were found. Thus, they suggested that the pointwise analysis between retinal sensitivity and inner retinal thickness is critical and is sensitive enough to depict early changes in the diabetic retina. Recently, this group reported thinning of the GCL and IPL detected by OCT in 134 patients with diabetes and no clinically evident diabetic retinopathy [42]. Although there was no difference in retinal sensitivity measured by fundus perimetry between patients with diabetes and healthy controls, a relationship between the GCL thickness and retinal sensitivity was observed. This result showed a close structure–function relationship in the very early stages of ganglion cell loss in diabetes. The size of stimuli used in the fundus perimetry test might be too large to delineate the slight ganglion cell loss and result in a negative difference between control and diabetic patients in retinal sensitivity.

The other group suggested that deep capillary plexus (DCP) change is the first sign of diabetic retinopathy [43]. They conducted a cross-sectional case-control study consisting of 34 patients with type 1 diabetes and 34 controls. They used multimodal imaging, including structural OCT, OCT angiography, and fundus perimetry. There was no significant difference in the macular thickness of the RNFL and GCC, the perfusion density of the superficial capillary plexus and choriocapillaris, the area of the foveal avascular zone, and the retinal sensitivity measured by MP-1. Only the perfusion density of the deep capillary plexus was significantly reduced in patients with diabetes (0.46 vs. 0.45). Nevertheless, neuronal change or vascular alteration preceding in the diabetic retina remains controversial.

### 2.3. Evaluation of Significant Diabetic Retinopathy

In 2010, Dunbar et al. reported the fixation evaluation by SLO and MP-1 in healthy controls and patients with significant diabetic retinopathy [44]. The stability of fixation is related to the central visual function. Rodenstock SLO (Rodenstock GmbH, Munich, Germany) is an established method for measuring fixation. The authors compared the fixation area of the bivariate contour ellipse area (BCEA) using Rodenstock SLO and fundus perimetry MP-1. In this study, there was no significant difference in the size of BCEA between normal subjects and those with diabetic retinopathy. However, other studies showed that fixation stability was less stable in eyes with diabetic retinopathy than in controls [45,46]. Fixation is more unstable in eyes with more advanced diabetic retinopathy, such as macular edema with deposition of hard exudate.

The relationship between retinal thickness and sensitivity has also been reported [47,48]. The correlation was not significant at all macular quadrants in eyes without macular edema but was significant in eyes with clinically significant macular edema. Okada et al. reported that retinal sensitivity in the central 2 and 10° of the macula was significantly associated with foveal retinal thickness in eyes with diabetic macular edema [48]. Using retinal mapping, the correlation between retinal sensitivity and thickness was strong in the superior and nasal quadrants and moderate in the central and temporal quadrants of the macula [47]. In those two studies, macular edema was evaluated using the retinal thickness measured by OCT, and retinal sensitivity was measured by MP-1 fundus perimetry.

The relationship between central visual acuity and retinal sensitivity measured by MP-1 and fixation stability was studied in 84 patients with diabetic retinopathy and significant macular edema in patients with type 2 diabetes [46]. There were significant differences between eyes with good fixation stability and poor fixation stability in the duration of diabetes, diabetic retinopathy, and retinal sensitivity within the central 2° and 8°, visual acuity, and central macular thickness. Eyes with eccentric fixation were more likely to be found in people with a longer duration of diabetes and chronic macular edema. In contrast, eyes with unstable fixation were more likely to be found in people with a shorter duration of diabetes. These data can be explained by establishing one location of the preferred retinal loci during the development of chronic macular edema.

In 2012, Soliman et al. studied macular retinal sensitivity in the subfield of the central 6 mm of the macular and retinal microstructures and measured the thickness by OCT [49]. They found a significant association between hard exudate and reduced retinal sensitivity, intraretinal cysts, and reduced retinal sensitivity. There was also a significant relationship between cysts in the inner and outer nuclear layers and reduced retinal sensitivity, but not in serous macular detachment. The authors suggested that the lack of significance between the subretinal fluid and retinal sensitivity was due to an underpowered study with relatively small samples (*n* = 20). In this study, the subfield outside the fovea was well tested using MP-1 fundus perimetry.

Vujosevic et al. reported no relationship between the types of macular edema, such as diffuse, focal, cystoid, and sponge-like, with or without subretinal fluid, and the retinal sensitivity measured by MP-1 fundus perimetry [45]. Instead, they found a significant relationship between the subfoveal hard exudate and impaired fixation stability and location. This study included 21 patients with type 1 diabetes and 77 patients with type 2 diabetes, with a mean age of 17 years. The deposition of hard exudate indicated degeneration of the photoreceptors and neuronal damage. The authors suggested that ophthalmologists should be aware of the eccentric fixation location determined by fundus perimetry before treating macular edema with foveal hard exudation using a focal laser.

## 3. Importance of the Functional Evaluation of Retinal Sensitivity in Clinical Trials for Diabetic Retinopathy

Randomized clinical trials are the gold standard for evaluating the efficacy of medical and surgical interventions. The primary outcome measure is usually visual acuity or the magnitude of visual acuity change. Visual acuity can be measured using the Snellen chart or Early Treatment Diabetic Retinopathy Study (ETDRS) chart and can be universally used. However, it is not enough to evaluate the patient’s visual function by visual acuity alone. The measurement of retinal sensitivity by fundus perimetry could be used in conjunction because it can evaluate point-to-point retinal function by measuring light sense, which is different from visual acuity (minimum recognizable acuity or legibility). Unfortunately, only a few studies have adopted a fundus perimetry test to measure outcomes [50,51,52,53].

Comyn et al. conducted a prospective, randomized clinical trial to compare the efficacy of intravitreal ranibizumab (22 eyes) with macular laser (11 eyes) for diabetic macular edema [50]. The primary outcome measures were the area of the foveal avascular zone, perifoveal capillary dropout grade, the presence of macular edema observed by OCT, visual acuity, contrast sensitivity, electroretinogram, and retinal sensitivity measured by MP-1 fundus perimetry. The visual acuity change showed a six letter gain in the ranibizumab group and a loss of 0.9 letters in the laser group at 48 weeks. The retinal sensitivity in the central 4 and 12° improved by 3.2 and 2.4 dB in the ranibizumab group and 1.9 and 1.1 dB in the laser group. The parameters of electrophysiological studies also improved significantly in the eyes treated with ranibizumab. The improvement in retinal sensitivity was negligible in the central 12°, which included a large area without retinal edema. Ranibizumab therapy, showed more improvement of visual function measured by multimodal test than macular laser therapy.

A phase 3 randomized controlled single-masked multicenter clinical trial was conducted by Sivaprasad et al. that aimed to test the clinical efficacy of light masks for treating non-central diabetic macular edema [51]. The light mask prevents retinal damage by reducing the dark current of rod photoreceptors. Visual acuity and retinal thickness were used as the primary outcomes. Retinal oximetry, multifocal electroretinography, and fundus perimetry were performed.

Gonzalez et al. conducted a study that included 46 patients from the DAVINCI study (a randomized, double-masked, phase 2 study) [52]. Patients were treated with a laser, intravitreal aflibercept (0.5 mg) every four weeks, 2 mg intravitreal aflibercept every eight weeks after three monthly doses, or 2 mg intravitreal aflibercept as needed after three monthly doses for 52 weeks. Retinal sensitivity was measured in the central and four inner ETDRS grids (five subfields) and four inner and four outer subfields (eight subfields). Retinal sensitivity was significantly better after intravitreal aflibercept treatment. The central sensitivity was comparable, but retinal sensitivity in the five subfields and eight surrounding subfields decreased at week 52 in the eyes treated with laser. Best-corrected visual acuity testing could not delineate this focal deterioration of retinal function, which may be a negative aspect of laser treatment.

The efficacy of yellow subthreshold laser treatment for diabetic macular edema was investigated by Chhablani et al. [53] in a prospective study including 30 eyes randomized into three groups. Ten eyes were treated with a subthreshold laser with a 5% duty cycle, ten eyes were treated by subthreshold laser with a 15% duty cycle, and the rest were treated with a conventional continuous wave laser. The primary outcome measures were central visual acuity and retinal sensitivity measured by fundus perimetry Maia at week 12. Retinal sensitivity in the central 10° was slightly reduced in the continuous laser group by −0.3 dB and increased in both subthreshold laser groups by 0.9 dB and 1.7 dB. Since there was no significant difference in visual acuity at week 12 among the three groups, the measurement of retinal sensitivity was a valuable modality to evaluate visual function.

## 4. Conclusions: The Importance of Fundus Perimetry in the Evaluation of Diabetic Retinopathy

The measurement of retinal sensitivity by fundus perimetry can reveal slight changes in eyes at an early stage of diabetic retinopathy when no morphological changes can be observed by ophthalmoscopy or OCT. Although this finding remains controversial, fundus perimetry appears to be a valuable tool for evaluating the preceding impairment of functional changes in the diabetic retina. Fundus perimetry has the great advantage of evaluating the point-to-point relationship between the retinal structure and function, especially in eyes with diabetic macular edema with a non-uniform spatial distribution. To evaluate the efficacy of treatments including laser and drugs, the measurement of retinal sensitivity and visual acuity is necessary because these two modalities evaluate different visual functions. The importance of fundus perimetry is undoubtedly increasing with numerous novel treatment modalities emerging for patients with diabetic retinopathy.

## Figures and Tables

**Figure 1 ijms-22-10726-f001:**
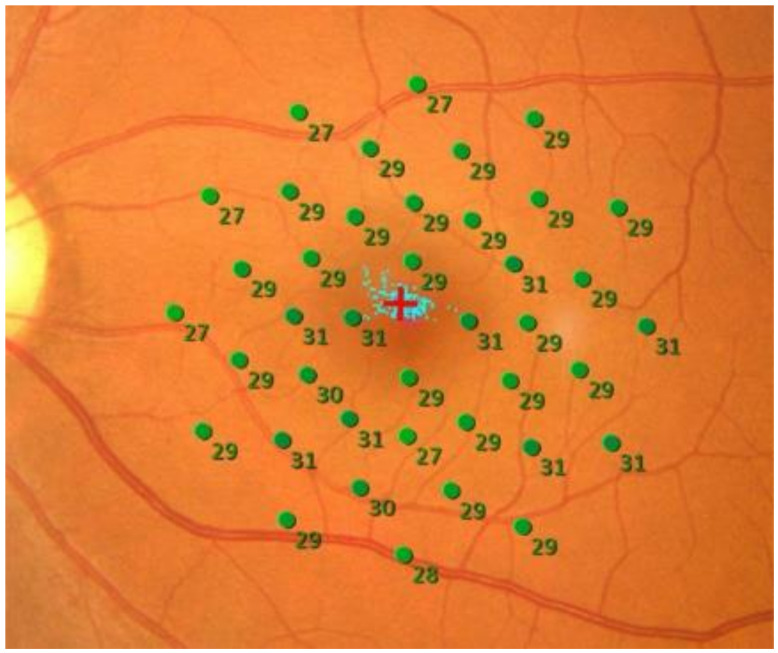
Retinal sensitivity in a normal eye. Retinal sensitivities measured by MP-3 were above 27 dB at all measurement points and are displayed in green. The 16 degrees—40 points pattern with background luminance 31.4 asb was used. The average retinal sensitivity in the central 20° is 28.5 dB.

**Figure 2 ijms-22-10726-f002:**
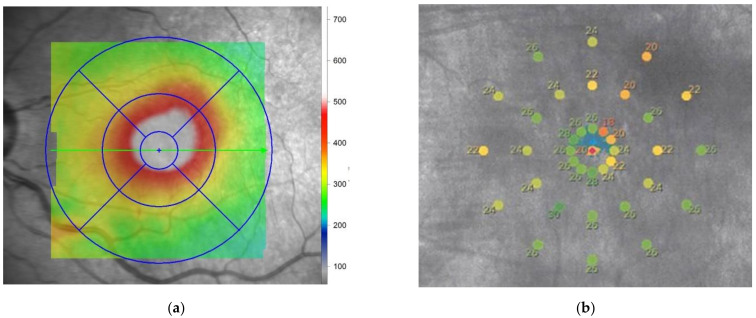
Retinal sensitivity in an eye with diabetic macular edema. (**a**) Retinal thickness map measured by optical coherence tomography. The thickness is increased at the superior temporal area to the central macula; (**b**) Mixed values of retinal sensitivity are presented in the macular area by maia. The average retinal sensitivity is 24.3 dB. Note that the low retinal sensitivity corresponds to the thick retina, suggesting retinal edema. The best-corrected visual acuity in the eye was 0.6.

**Table 1 ijms-22-10726-t001:** Comparison of different types of fundus perimetry.

	MP-1	Maia	MP-3
Fundus image	Infrared	SLO	SLO
Measurement area	40° circle	20° × 20°	40° circle
Stimulus size	Goldmann I–V	Goldmann III	Goldmann I–V
Smallest pupil size	4 mm	2.5 mm	4 mm
Background luminance	4 asb	4 asb	4 asb/31.4 asb
Display	LCD	LED	LED
Maximum stimuli	400 asb	1000 asb	10000 asb
Dynamic range	0–20 dB	0–36 dB	0–34 dB
Alignment	Manual	Manual	Auto
Eye tracking	25 Hz	25 Hz	30 Hz
Fundus photo	Color	Grayscale	Color
Capture angle	45° circle	36° × 36°	45° circle
Resolution of image	1392 × 1024 pix	1024 × 1024 pix	4016 × 3008 pix
Focus range	Manual: −15 to +15D	Auto: −15D to +10D	Auto: −12D to +15D; Manual: −25D to +15D
Scotopic devices	MP–1S	S–Maia	n/a

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
