# Peer review of "Detecting Diabetic Retinal Neuropathy Using Fundus Perimetry"

_ijms, 2021, doi:10.3390/ijms221910726_

Round 1

Reviewer 1 Report

This is generally very interesting and well written review paper although I am not sure if it is well suited for the journal focused on the molecular medicine. The main conclusion is that fundus microperimetry appears to be a valuable tool for evaluating the  preceding impairment of functional changes in the diabetic retina. Only clinically relevant findings beneficial for ophthalmologists are included in this article, thus it would be better to submit it to ophthalmological or multidisciplinary journal.

Author Response

This is generally very interesting and well written review paper although I am not sure if it is well suited for the journal focused on the molecular medicine. The main conclusion is that fundus microperimetry appears to be a valuable tool for evaluating the preceding impairment of functional changes in the diabetic retina. Only clinically relevant findings beneficial for ophthalmologists are included in this article, thus it would be better to submit it to ophthalmological or multidisciplinary journal.

Answer:  I appreciate the reviewer’s comment. As this reviewer mentioned, most of this review is associated with clinical findings using fundus perimetry. On the other hand, this review includes information about the fundamental understanding of the visual function and testing the visual field. The latter part of this review also included the topic about early changes of the neural retina in which the neural damage may proceed to vascular damage. This change can be detected by fundus microperimetry, and I think this information must be beneficial not only for clinicians but also for basic researchers who are interested in the diabetic retina.

Reviewer 2 Report

Baba has reviewed the role of microperimetry in evaluating retinal function in diabetic patients. The manuscript provides evidence supporting the usefulness of microperimetry in detecting early functional changes and its potential application as a trial endpoint in diabetic patients. Below are some comments and suggestions for the author.

General comments:

  1. The title can be misleading for general readers. The term “Diabetic Neuropathy” is generally used for peripheral neuropathy. I appreciate that the neurosensory retina is part of the peripheral nervous system; however, I would recommend using either “Diabetic Retinal Neuropathy” for further clarification, or “Diabetic Retinopathy” which is a general and accepted description of diabetic retinal complications and has been used by the author throughout the manuscript.

  1. Alternative terms for microperimetry (MP) include “fundus perimetry”, “fundus-controlled perimetry (FCP)” and “fundus-driven Perimetry”. I recommend consistent use of one of these terms, rather than “fundus microperimetry”, throughout the manuscript, including the title.

  1. In perimetry (including microperimetry), the term “stimulus” is usually used instead of “optotype”. Optotype is usually reserved for shapes and letters used in visual acuity testing, and less commonly, Goldmann perimetry. Please consider revising throughout the manuscript.

Specific comments:

  1. Lines 18-20: The sentence structure needs revision.

  1. Lines 31-32: Perimetry devices measure retinal sensitivity based on the “differential light sensitivity”, which is defined as the difference between the background luminance and stimulus intensity. Hence, light perception and retinal sensitivity cannot be used interchangeably.

  1. Lines 37-38: Please mention that the blind spot is located temporal to the visual field center.

  1. Line 49: The readers might be more familiar with “arcmin” or “minutes of arc” rather than its sign in (3’).

  1. Lines 53-59: similar to lines 31-32. Retinal sensitivity is measured by the differential light sensitivity, rather than light perception.

  1. Lines 87-97: Again, please clarify that the device measures the differential light sensitivity, which depends on the background luminance. Also, not all microperimeters provide color fundus image (lines 97-97).

  1. Figure 1: Please indicate the name of the microperimetry machine and testing grid.

  1. Lines 102-115: Despite real-time high-speed eye-tracking, localization of the stimulated retinal area may not be precise. Please briefly discuss the sources, magnitude, and significance of this alignment error.

  1. Line 120: Please define maia (MAcular Integrity Assessment). In addition, “Centerview” should be “Centervue”.

  1. Lines 123-125: In fact, the everyday life condition is photopic condition, which is tested at a background luminance level of 31.4 asb (or 10 cd/m2). Mesopic vision is tested under low light levels at a background luminance of 4 asb (1.27 cd/m²). By the way, it is correct that measuring retinal sensitivity in everyday life condition is an advantage of MP-3.

  1. Table 1: it is useful to list scotopic devices (MP-1S and S-MAIA) as they can measure rod-mediated sensitivity, which might be affected even earlier than mesopic or photopic sensitivity.

  1. Lines 128-129: similar to lines 123-125.

  1. Line 130: The correct company name is “Centervue”.

  1. Line 140: What is the reference for the normal values? Please also provide the normal values for MP-1 and MAIA as studies using these devices are included in sections 2 and 3.

  1. Lines 194+: Please avoid presenting excessive details on the previous works (e.g. standard deviations, decimals of age and foveal thickness, second decimals of sensitivity) and only include relevant and important numbers. Please further highlight the main conclusion of each study rather than detailed results. Details of other modalities and outcome measures are unnecessary unless they highlight the advantage/limitation of microperimetry compared with other techniques.

  1. Lines 350-361: Please provide the type of microperimetry device used in this study.

  1. Lines 362-367: Please indicate what microperimetry device was used and what was the conclusion?

  1. Lines 368-380: Again, please provide the type of microperimetry machine.

  1. Line 387: Manufacturer information is not necessary after the first instance (depends on the journal policy).

Author Response

Reviewer 2

Baba has reviewed the role of microperimetry in evaluating retinal function in diabetic patients. The manuscript provides evidence supporting the usefulness of microperimetry in detecting early functional changes and its potential application as a trial endpoint in diabetic patients. Below are some comments and suggestions for the author.

General comments:

The title can be misleading for general readers. The term “Diabetic Neuropathy” is generally used for peripheral neuropathy. I appreciate that the neurosensory retina is part of the peripheral nervous system; however, I would recommend using either “Diabetic Retinal Neuropathy” for further clarification, or “Diabetic Retinopathy” which is a general and accepted description of diabetic retinal complications and has been used by the author throughout the manuscript.

Answer: I changed the title to “Detecting Diabetic Retinal Neuropathy Using Fundus Perimetry”. (Line 1-3)

Alternative terms for microperimetry (MP) include “fundus perimetry”, “fundus-controlled perimetry (FCP)” and “fundus-driven Perimetry”. I recommend consistent use of one of these terms, rather than “fundus microperimetry”, throughout the manuscript, including the title.

Answer: I changed to “fundus perimetry” throughout the manuscript.

In perimetry (including microperimetry), the term “stimulus” is usually used instead of “optotype”. Optotype is usually reserved for shapes and letters used in visual acuity testing, and less commonly, Goldmann perimetry. Please consider revising throughout the manuscript.

Answer: I changed “optotype” to “stimulus” throughout the manuscript.

Specific comments:

Lines 18-20: The sentence structure needs revision.

Answer: I changed the sentence to “In this review, the importance of measuring retinal sensitivity to evaluate visual function in the early stages of diabetic retinopathy was discussed. The usefulness of retinal sensitivity as an outcome measure in clinical trials for treatment modalities is also presented.” (Line 19-21)

Lines 31-32: Perimetry devices measure retinal sensitivity based on the “differential light sensitivity”, which is defined as the difference between the background luminance and stimulus intensity. Hence, light perception and retinal sensitivity cannot be used interchangeably.

Answer: I changed the sentence to “Retinal sensitivity is the way we perceive light, and the visual field is an accumulation of regions with various retinal sensitivities which is measured by the difference between the background luminance and stimulus intensity.” (Line 33-35)

Lines 37-38: Please mention that the blind spot is located temporal to the visual field center.

Answer: I changed the sentence to “At a distance of approximately 15° temporal from the center, there is an area with zero retinal sensitivity called the blind spot that is approximately 5° in diameter.” (Line 40-42)

Line 49: The readers might be more familiar with “arcmin” or “minutes of arc” rather than its sign in (3’).

Answer: I changed it to “three minutes of arc” for better understanding. (Line 53-54)

Lines 53-59: similar to lines 31-32. Retinal sensitivity is measured by the differential light sensitivity, rather than light perception.

Answer: I changed the first line in this paragraph to “Retinal sensitivity, measured in visual field testing, is the ability to perceive the difference between the background luminance and stimulus intensity.” I also changed the third sentence to “In other words, if the examinee can perceive dark stimulus, the examinee has high retinal sensitivity and high visual function.” (Line 57-58, 61-62)

Lines 87-97: Again, please clarify that the device measures the differential light sensitivity, which depends on the background luminance. Also, not all microperimeters provide color fundus image (lines 97-97).

Answer: I changed to “By gradually changing the brightness of the stimulus from bright to dark, the darkest luminance (threshold) at which the stimulus can be differentiated from the background luminance is determined and is used as the retinal sensitivity. [11] In addition, I removed “color” from the last sentence in the paragraph. (Line 102-105, 109)

Figure 1: Please indicate the name of the microperimetry machine and testing grid.

Answer: I added “The 16 degrees – 40 points pattern with background luminance 31.4 asb was used.” to the legend of Figure 1. (Line 114)

Lines 102-115: Despite real-time high-speed eye-tracking, localization of the stimulated retinal area may not be precise. Please briefly discuss the sources, magnitude, and significance of this alignment error.

Answer: I added “The localization of the stimulated retinal area may not precise throughout the session because the fixation is not always stable. The mean eye tracking accuracy of 4.9 minutes of arc has been reported by MP-1.[10] The tracking frequency is one of factors which varies from device to device. In addition, the patients with macular disease have poorer fixation stability with less accurate localization of stimulus. The head movements as well as the eye movement also affect the alignment during the test.” (Line 97-102)

Line 120: Please define maia (MAcular Integrity Assessment). In addition, “Centerview” should be “Centervue”.

Answer: I defined maia as “MAcular Integrity Assessment”, and corrected the company’s name as Centervue. (Line 137)

Lines 123-125: In fact, the everyday life condition is photopic condition, which is tested at a background luminance level of 31.4 asb (or 10 cd/m2). Mesopic vision is tested under low light levels at a background luminance of 4 asb (1.27 cd/m²). By the way, it is correct that measuring retinal sensitivity in everyday life condition is an advantage of MP-3.

Answer: Only MP-3 has the background luminance level of 31.4 asb and has advantage to measure retinal sensitivity under the conditions close to everyday life. I changed “mesopic” to “photopic”. (Line 141)

Table 1: it is useful to list scotopic devices (MP-1S and S-MAIA) as they can measure rod-mediated sensitivity, which might be affected even earlier than mesopic or photopic sensitivity.

Answer: I added MP-1S and S-MAIA in Table 1 as scotopic version of MP-1 and MAIA. (Line 145)

Lines 128-129: similar to lines 123-125.

Answer: Answer: Only MP-3 has the background luminance level of 31.4 asb and has advantage to measure retinal sensitivity under the conditions close to everyday life. I changed “mesopic” to “photopic”. (Line 149-150)

Line 130: The correct company name is “Centervue”.

Answer: I changed it to “Centervue”. Thank you. (Line 151)

Line 140: What is the reference for the normal values? Please also provide the normal values for MP-1 and MAIA as studies using these devices are included in sections 2 and 3.

Answer: I added “The normal retinal sensitivity ranges from 24 to 36 dB in Maia, and from 12 to 20 dB in MP-1.” (Line 163-164) I also added normal values along with the values of retinal sensitivity shown in the section 2 and 3.

Lines 194+: Please avoid presenting excessive details on the previous works (e.g. standard deviations, decimals of age and foveal thickness, second decimals of sensitivity) and only include relevant and important numbers. Please further highlight the main conclusion of each study rather than detailed results. Details of other modalities and outcome measures are unnecessary unless they highlight the advantage/limitation of microperimetry compared with other techniques.

Answer: I removed the excessive digits from data in the previous works. I also made the description of previous work more concise.

Lines 350-361: Please provide the type of microperimetry device used in this study.

Answer: I added the type of fundus perimetry device. (Line 341)

Lines 362-367: Please indicate what microperimetry device was used and what was the conclusion?

Answer: I added the type of fundus perimetry device. I also added the conclusion as “Ranibizumab therapy showed more improvement of visual function measured by multimodal test than macular laser therapy.” (Line 375-376)

Lines 368-380: Again, please provide the type of microperimetry machine.

Answer: The type of fundus perimetry device was not mentioned in this study by Sivaprasad et al.

Line 387: Manufacturer information is not necessary after the first instance (depends on the journal policy).

Answer: I removed the manufacturer information of maia. (L400)

Reviewer 3 Report

The manuscript is well written, although no materials and methods section is provided. Systematic review are preferable to narrative ones. Please improve it if it is possible.

Author Response

Reviewer 3

The manuscript is well written, although no materials and methods section is provided. Systematic review are preferable to narrative ones. Please improve it if it is possible.

Answer: I appreciated the reviewer’s suggestion. I agree with the comment that a systematic review is better than a narrative one. I wrote this review not only for clinical ophthalmologists but also for the basic researchers in the field of diabetic retina. Therefore, I included basic information on visual function and visual field testing in most of the first part. I performed Pubmed research and covered most of the recent work using fundus perimetry but it was not completely systematic because of the above reason. I still believe this narrative review is informative on the recent topics about the use of microperimetry for evaluating diabetic retinopathy.

Round 2

Reviewer 1 Report

I still believe that this paper is dedicated more for clinicians rather than for researchers, as the physiological background is not extensively investigated. As the journal has impact factor of 5.9, for me this manuscript has no such scientific impact. Moreover, the author has not published papers in this area (perimetry) and is not an expert in this field. I am not sure if the term “diabetic retinal neuropathy” exists. The term “Diabetic neuropathy” refers to optic nerve and the term “peripheral diabetic neuropathy” refers to  peripheral nerves. The retina should be considered as a vascularized neural tissue and during diabetes both microvasculopathy and neurodegeneration of the retina occur. In the present article the detection of diabetic retinopathy (macular edema, proliferative diabetic neuropathy) using fundus perimetry is described in many studies, rather detection of neuropathy of the optic nerve.

Reviewer 2 Report

The author has responded to all my comments accordingly and I am satisfied with the revised version of the manuscript.

Reviewer 3 Report

The author did not improved the manuscript as requested. In my opinion, systematic review are mandatory, especially if the aim is inform basic researchers.